# Long-Term Effects of COVID-19 Pandemic on Migraine in Adolescents. A Retrospective Analysis of the Population Attending the Headache Center in Different Phases of the Pandemic

**DOI:** 10.3390/brainsci13020273

**Published:** 2023-02-06

**Authors:** Martina Proietti Checchi, Samuela Tarantino, Fabiana Ursitti, Gabriele Monte, Romina Moavero, Giorgia Sforza, Michela Ada Noris Ferilli, Teresa Grimaldi Capitello, Federico Vigevano, Massimiliano Valeriani, Laura Papetti

**Affiliations:** 1Developmental Neurology, Bambino Gesù Children Hospital, IRCCS, 00165 Rome, Italy; 2Child Neurology and Psychiatry Unit, Tor Vergata University of Rome, 9220 Rome, Italy; 3Unit of Clinical Psychology, Bambino Gesù Children Hospital, IRCCS, 00165 Rome, Italy; 4Neurological Sciences and Rehabilitation Medicine, Bambino Gesù Children Hospital, IRCCS, 00165 Rome, Italy; 5Center for Sensory-Motor Interaction, Aalborg University, 9220 Aalborg, Denmark

**Keywords:** migraine, COVID-19, lockdown, adolescents, anxiety, depression

## Abstract

Background: Literature data report that the first COVID-19 pandemic had an impact on the progression of migraine both in adults and children. The present study aimed to verify how the migraine course and psychological aspects varied in adolescent patients in relation to some of the different phases of the COVID-19 pandemic and compared with the months before COVID-19. In addition, the relationship between the characteristics of headache episodes and psychological and school-related aspects were analyzed. Methods: The study included 418 adolescents. Based on the timing of the evaluation, they were categorized into patients observed before the COVID-19 pandemic (pre COVID) or during the first (COVID 1) or second (COVID 2) wave of the pandemic. Subjects were also categorized into three further groups: those who had high or low frequency of migraine attacks during the month, those who had mild or severe pain during the attack, and those who were taking prophylactic drugs. The Patient Health Questionnaire-9 (PHQ-9) and General Anxiety Disorder-7 (GAD-7) scales were utilized to assess depression and anxiety. Results: We observed a significant increase in the frequency of attacks and the use of prophylactic drugs during the COVID 2 period compared to the COVID 1 and pre-COVID periods (*p* < 0.05). Patients showed higher levels of anxiety and depression during each of the two COVID periods compared with the pre-COVID months (*p* < 0.05), especially during the COVID 2 period (*p* < 0.05). Conclusion: Our results show long-term negative impacts of the COVID-19 pandemic on clinical parameters and psychological symptoms in adolescents with migraine.

## 1. Introduction

The COVID-19 pandemic affected people all over the world. To contain the spread of the virus, governments worldwide enforced extraordinary and strict forms of public health interventions and restrictions, including physical distancing, lockdowns, and quarantines, leading to closure of business and sports facilities. For children and adolescents, the closure of schools, the introduction of new forms of learning such as online lessons, and the suspension of extracurricular activities represented important changes, leading to social isolation [1,2]. Although these restrictive measures helped to prevent and control the rapid rise of COVID-19 and have now been lifted, the community continues to deal with the virus. The dramatic changes in people’s lifestyles and the loss of daily routine, together with the fear of infection, have led to an increased rate of anxiety, depression, and feelings of loneliness [3,4,5,6,7]. Despite the higher risk of mortality among the older population, the COVID-19 pandemic impacted mostly young people’s mental health [8,9,10,11]. Heterogeneous effects on the mental health of children and adolescents have been described. Internalizing symptoms such as anxiety and depression were associated with the first wave of the pandemic (March to May 2020), and an increase in suicidal attempts or ideation among adolescents during the second wave (September to December 2020) was reported [12].

Increased family stress, less time spent with peers, difficulties participating in online learning, and suboptimal eating and sleeping habits [13,14] were the most common factors impacting children’s and adolescents’ mental health during the pandemic. The course of chronic pain conditions such as migraine was influenced by psychological stress [15,16,17].

Stress, negative life events, and dysfunctional family relationships may play important roles as risk factors for migraine chronicization [14,15,16,17,18]. However, despite the negative impact of the first wave of the COVID-19 pandemic on patients’ lives, together with increased levels of stress, concerns about infection, and changes in sleep, an unexpected improvement in headaches were described both in adults and children [19,20,21,22]. Differing results have been reported regarding the long-term effects of the pandemic on migraine severity. In adults, the extension of restrictive measures, the insecurity associated with the outcome of the pandemic, and the associated negative emotions resulted in a worsening of migraine severity.

To the best of our knowledge, only two studies have to date explored the impact of the pandemic on mental health and migraine features in adolescents, and they did not explore the long-term effects of the pandemic nor use specific standardized tools to assess psychological symptoms [19,20]. This current retrospective study aims to investigate the impact of COVID 19 and the long-term effects of the pandemic on migraine characteristics and psychological symptoms.

Our main aims were: (1) to explore differences in migraine characteristics between the pre-COVID and COVID-19 periods; (2) to investigate emotional and school-related aspects affecting subjects in the different periods; (3) to study the relationship between characteristics of headache episodes and psychological and educational aspects.

## 2. Materials and Methods

### 2.1. Selection and Classification of the Patients

Patients were identified through a systematic review of clinical records of adolescents referred to the Headache Center, Division of Neurology, of the Bambino Gesù Children’s Hospital in Rome, from September 2018 to January 2022.

Data on the clinical characteristics of headaches, the therapy administered, and the diagnosis at discharge were collected in a headache diary given at the first consultation and returned by the family at the second consultation. Every patient underwent a neurological examination. Only the patients with a diagnosis of migraine, with or without aura, according to the criteria of the International Classification of Headache Disorder, Third Version (ICHD-3) [23] were included. Tension-type headaches, trigeminal autonomic cephalalgias, secondary headaches, or patients suffering from any other neurological disease were excluded from our study.

According to the date of referral, patients were grouped into “pre-COVID” (PC, from September 2018 to February 2020) or “COVID” (C, from March 2020 to January 2022). The “COVID” group was further divided into “COVID 1” (C1, from March to October 2020, characterized by lockdown, the start of remote video lessons, and summer holidays) and “COVID 2” (C2, from November 2020 to January 2022, characterized by return to school and/or remote video lessons, summer holidays, and the extension of COVID-19 restrictions) (Figure 1).

According to their frequency, patients were classified into two groups: (1) high frequency (HF; participants complaining of more than 4 attacks per month); (2) low frequency (LF; adolescents with 4 or fewer episodes per month). A cut-off of 4 attacks per month was established because this is generally the frequency above which to consider starting migraine prophylaxis in childhood [24,25].

The delimitation point was chosen for three reasons: (1) the numbers of patients suffering from chronic and intermediate frequency attacks were too low to allow reliable statistical comparison; (2) a simple distinction between chronic and episodic patients would have resulted in including non-chronic patients with a high frequency of attacks in the same group as individuals with a very low frequency of attacks; (3) the chosen demarcation point was rationalized to distinguish patients who needed prophylactic treatment from those who did not.

In addition, the average number of migraine episodes in the previous two months was assessed. Pain intensity was classified into severe (SI) and mild (MI) based on interference with daily activities. Patients were classified into those who were receiving prophylaxis treatment (PT) (with treatment terminated before 4 weeks) and those who were not (NT). In addition, subjects were divided according to the time of year in which they were evaluated at the Day Hospital of the Headache Center (school term or summer). School-related information, including the grade of school attended, type of teaching, and school performance, were collected by psychological interview.

The type of teaching was divided into in-person and remote teaching, while students’ performance was classified into two groups: good/very good (grades 7–10) and sufficient/insufficient (grades 0–6). The psychological screening was performed by psychologists (MPC and ST) with expertise in pediatric psychological evaluation. The psychological tests were administered in a single sitting. All patients were required to be able to read, understand, and answer every item on the questionnaires.

The Institutional Review Board of Bambino Gesù Children Hospital provided approval for this study. 

Written informed consent for patient information to be published was provided by the parents of subjects involved in the research.

### 2.2. Instruments

Anxiety was evaluated using the General Anxiety Disorder-7 (GAD-7) scale; depression was estimated using the Patient Health Questionnaire-9 (PHQ-9) scale.

#### 2.2.1. Anxiety

The GAD-7 is a brief, 7-item, self-reported measure of anxiety, rated on a 4-point Likert scale [26]. Four alternatives are offered: 1: not at all; 2: some days; 3: more than half of days; and 4: almost every day. The total score was categorized into “no anxiety” (range: 0–4), “mild anxiety” (range: 5–9), “moderate anxiety” (range: 10–14), and “severe anxiety” (equal to 15). 

#### 2.2.2. Depression

The PHQ-9 is a brief, self-administered measure of depressive symptoms, with nine items that fit the diagnostic criteria for major depressive disorders. Item 9 investigates the idea of harming oneself or wanting to die. The scale is a screening instrument, and thus does not itself provide a clinical diagnosis. Symptoms are rated using a 4-point scale (0 = not at all; 1 = some days; 2 = more than half of days; and 3 = almost every day) regarding the subject’s experience in the previous two weeks. The severity of depressive symptoms was categorized as “no depression” (range: 0–4), “mild depression” (range: 5–9), “moderate depression” (range: 10–14), and “severe depression” (equal to 15) [27,28].

### 2.3. Statistical Analysis

The statistical analysis of the data was conducted using the software Jamovi and R (the R Foundation for Statistical Computing).

For the primary objective of analyzing and comparing the course of headache and psychological symptoms during the PC and C periods, we considered three primary endpoints.

The first endpoint was to analyze headache trends such as the frequency of headache episodes, which may have been improved, unchanged, or worsened, and the use of prophylactic treatment. The second endpoint was to analyze the results of the psychological questionnaires, educational performance, and type of teaching. The third endpoint was to identify the relationship between the variables analyzed in the first endpoint and those in the second endpoint.

First, descriptive statistics were used to describe the participants’ characteristics, including socio-demographic data, psychometric instrument scores, and headache characteristics. Subsequently, the demographic and headache characteristics of the patients were compared between the PC and C groups using Pearson’s chi-squared test. Continuous variables were represented by the mean standard deviation, while categorical variables were represented by absolute frequencies and percentages. Pearson’s correlation coefficient was used for comparison between continuous variables where appropriate. Continuous variables were analyzed using the Shapiro–Wilk test to assess the normal distribution. Student’s t test for independent samples was applied for comparison between categorical variables with two levels and continuous variables, as appropriate.

The ANOVA test was applied to compare the psychometric test scores between patients in the PC and C groups. For all statistical tests, we set statistical significance at *p* < 0.05. 

## 3. Results

We studied 418 adolescents with migraine with and without aura (32 subjects with aura and 386 subjects without aura; mean age 14 ± 1.7 years; 110 males and 308 females). In the PC period, 118 patients were evaluated, while 300 patients were evaluated in the C period. The C group was composed of 63 patients evaluated in the C1 period and 237 patients evaluated in the C2 period (Table 1).

### 3.1. Migraine Features

Of the total sample, 49% showed attacks with HF, while 51% presented LF. There were no significant differences in the percentages of patients with HF and LF attacks between the PC and total C groups (*p* = 0.262). The PC patients reported an overall average of 5.75 episodes of attacks in the last two months, while the C patients had a mean of 5.90 (C1 = 4.27, C2 = 6.33) (Figure 2 and Figure 3). 

When we compared the frequency of attacks between the PC, C1, and C2 groups, we found that more attacks were experienced by patients with HF in the C2 group (56.1% vs. 44.9%; *p* = 0.046) and C1 group (56.1% vs. 31.7%; *p* < 0.001) (Table 1).

When we analyzed the variation of migraine frequency in the PC, C1, and C2 groups during the months with school activities vs. summer holidays, for all three groups we found improvements in the trends of headache frequency during the summer. In the C2 period, the variation among patients with HF was significant, with a lower frequency of patients with HF attacks during the summer holidays (35%), compared with months that included school activities (60%; *p* < 0.05). The percentage of HF patients was significantly higher during the C2 period (60%) compared with PC (46%), and C1 (31%; *p* < 0.01).

In the periods analyzed, PT was observed more frequently in the C period (PC = 26.3% vs. C = 38%, *p* < 0.05), particularly in the C2 period (PC = 26.3% vs. C2 = 42%, *p* = 0.017; C1 = 25% vs. C2 = 42%, *p* < 0.05) (Table 1). In relation to seasonal period, we found a significant increase in PT during the school months of C2 compared with PC (43% vs. 24%, *p* < 0.01), whereas no significant differences were found in the summer holidays between all groups (*p* = 0.48).

There were no significant differences in the intensity of headache episodes between all groups (*p* = 0.848) (Table 1).

### 3.2. Psychological Findings

#### 3.2.1. Anxiety

There was an increase in patients with severe scores in the anxiety test in the C period (17%) compared with the PC period (8%) (*p* < 0.05). Specifically, we observed higher scores on anxiety tests in the period C2 compared with PC and C1 (mean values of 8.8 vs. 7.55 vs. 8.7, respectively; *p* < 0.05) (Figure 4). Our data show a higher presence of anxiety symptoms in patients with HF (mean value: HF 9.42, LF 7.55, *p* < 0.001). We found higher anxiety scores in patients with HF in the C period than in the PC period (mean value 9.79 vs. 8.23, *p* < 0.05).

Considering the anxiety scores of the subjects evaluated during the months with school activities vs. summer holidays, we did not find significant differences in the overall sample (mean value 8.65 vs. 7.78; *p* = 0.129), but higher anxiety scores emerged in the PC period in the months with school activities compared with the summer holidays (mean value 8.15 vs. 4.60; *p* < 0.01). Finally, no significant differences in anxiety symptoms were observed in patients who received in-person instruction compared with those who received remote instruction (mean value 8.29 vs. 9.39; *p* = 0.06). In patients that received in-person teaching, there were differences in anxiety symptoms between the PC and C periods (mean value 7.65 vs. 8.78; *p* < 0.05).

#### 3.2.2. Depression

There was a statistically significant difference between the PC and C periods regarding depressive symptoms (mean value 7.63 vs. 9.67, *p* < 0.01) (Figure 5). The mean scores in tests exploring depression were 7.63 in PC period, 9.37 in C1, and 9.75 in C2. The differences between C1 and C2 were not significant (*p* = 0.571). In the C period, we observed an increase in patients with “moderate” (38.7%) or “severe” (16.7%) depression compared with the PC period (20.3% moderate, 7.6% severe) (*p* < 0.05). In general, our data reveal a greater presence of depressive symptoms in patients with HF (mean HF value 9.95, LF 8.27, *p* < 0.01). We found higher depressive scores in patients with HF in the C2 period than in the PC period (mean value 10.58 vs. 8.45, *p* < 0.01).

Considering the depressive scores of subjects evaluated during the months with school activities vs. summer holidays, we did not find significant differences (mean value 9.11 vs. 9.01; *p* = 0.855). There were no differences in depressive symptoms between in-person and remote teaching (mean value 8.91 vs. 9.92, *p* = 0.084). Among patients that received in-person teaching, there were differences in depressive symptoms between the PC and C periods (mean value 7.63 vs. 9.92; *p* < 0.01).

Furthermore, in-person teaching showed an association with increased frequency of attacks and depressive levels in the C period compared with the PC period (*p* < 0.05). No differences were observed between C1 and C2 in terms of the effects of distance or face-to-face teaching (*p* = 0.213). 

## 4. Discussion

Our study analyzed the long-term impact of the COVID-19 pandemic on migraine and psychological symptoms in adolescents. The main findings are: (1) there was a tendency towards migraine improvement during the first wave of COVID-19 (C1), followed by an increase in headache attacks and use of prophylaxis during the second wave (C2); (2) in all three periods analyzed, the frequency of migraine attacks and the use of prophylactic drugs increased further during the months with school activity compared with the months of summer vacation; (3) higher scores were reported for anxiety and depression tests during the period C2 compared with the period C1.

Few studies have explored the effects of the COVID-19 pandemic on headache outcomes, and most of these have focused on the first wave of the pandemic (lockdown). Studies on adults showed that lockdown and related changes in lifestyle had a beneficial effect on migraine symptoms [29]. An Italian study that considered the pandemic’s long-term effects on headaches found that the positive effect on migraine of the first lockdown period reverted to a negative outcome during the second lockdown period. The study showed that in patients with episodic migraine the frequency of episodes not only returned to the basal condition but even worsened [30].

Data on the effects of the COVID-19 pandemic on pediatric headache are sparse [19,20,31]. In an original Italian study involving 707 adolescents diagnosed with headaches, Papetti et al. described a positive effect of lifestyle changes such as lockdown on the frequency and intensity of pediatric migraine, regardless of pharmacological prophylaxis and geographic area of origin. An improvement in migraine symptoms during lockdown was confirmed in a study showing a reduction in intensity and frequency of attacks during the first wave of COVID-19 [20]. The authors reported a clinical worsening of migraine frequency in patients that reported higher anxiety during the lockdown phase. With the continuation of the pandemic and the prolongation of quarantine, children and adolescents generally reported increased emotional distress and a worse quality of life, due to social isolation and limitations on activity [31].

Another study analyzed the effects of the pandemic on headaches in 107 pediatric subjects in the period from summer 2020 to winter 2021; however, without specifying the exact dates of the start and end of the observation. The authors found an increased frequency of chronic headaches from 40% (N = 42) to 50% (N = 54), and an increase in constant daily headaches from 22% (N = 24) to 36% (N = 38). Patients reported worsened anxiety (*n* = 58, 54%), mood (*n* = 50, 47%), and workload (*n* = 49, 46%). However, that study had numerous limitations, including the use of an interview and the failure to analyze the correlation between the severity of the migraine and the time period in which the questionnaire was completed. This is a crucial element because migraine is a disease with a seasonal pattern, especially in pediatric age groups, and the environmental measures of prevention, restriction, and organization affecting children of school age were very different during the various phases of the pandemic [32].

Our study revealed that changes in the frequency of attacks were correlated with multiple factors that could have had an effect on the disease, and those factors varied over the time periods considered. Variations in headaches were considered with respect to the lockdown measures in force at the time, the presence or absence of school activities, the type of school activities, and the coexistence of anxiety and depression.

We found there had been an improvement in the frequency of migraines during the first phase of the pandemic. In this period, the main environmental changes were the suspension of school activities, the possibility of remote lessons, and the complete lockdown, in a context in which it was not yet known what the economic and social repercussions of the pandemic would be.

It should be emphasized that only patients with a comparatively poor clinical picture tended to visit hospital during the lockdown period. This may explain why, although there was an improvement in headache frequency in this period compared with the pre-pandemic period, the difference was not statistically significant compared with our previous study administered online for the entire population followed for headache, including those who did not attend a return visit [19]. Probably, this improvement in headache is attributable to a reduction in school stress and other lifestyle changes associated with lockdown.

Meanwhile, our data showed an increase in headache frequency and an increased use of prophylaxis during the second wave of the pandemic. This was the period for which the worst migraine effects were reported, during which there was a resumption of school activities [either electronically or in person] combined with another set of restriction measures, albeit less than in phase C1. Probably this comparative worsening was due both to the resumption of school as a source of stress and to the renewal of the restriction measures, which brought the awareness of not having emerged from the emergency of the COVID-19 pandemic.

The role of school stress as a determinant of headache worsening is also suggested by the variation in headache frequency when comparing the months of school activities and the months of summer holidays. In all three periods considered (PC, C1, and C2), we observed a reduction in the frequency of attacks and in the intake of prophylactic therapies during the summer months.

The school environment can be a challenge for young people with migraine, for several reasons, including the fixed hours, the need to stay focused, the physical difficulties of sitting for an entire class period, and external or internal demands for high performance in the school setting. After a gap of about six months caused by the first COVID-19 pandemic emergency, children and adolescents had to return to school, alternating between face-to-face teaching and online lessons, and adapting to new rules and precautions against the virus. These conditions could have resulted in increased stress, which is known to have a negative impact on migraine. These factors related to school stress are likely to have impacted migraine severity during the second wave, together with the persistence of restrictive measures enforced to prevent the second wave of COVID-19 infection.

We also observed higher scores on tests for anxiety and depression in period C2 compared with periods PC and C1. These results, showing high levels of anxiety and depression, especially during the second wave, confirm the previous literature on this topic [30,32,33]. We hypothesize that, while a resilient mechanism may have assisted in coping with rapid lifestyle changes and pandemic stress in the first wave of the pandemic, leading to better outcomes in terms of headache [32], children and adolescents generally reported increased distress and negative emotional reactions associated with the continuation of factors such as prolonged COVID-19 restrictions and uncertainty about the pandemic outcome. Previous data suggest that the large diffusion of infection and the perpetuation of the health emergency linked to the COVID-19 pandemic had an overall unfavorable effect on migraine in the adult population [30]. This variation with respect to the first pandemic wave was connected with negative emotional reactions such as anger, anxiety, fear, and risk perception [33]. In the first pandemic phase, the limited impact of emotional behavior on migraine may have been due to a phenomenon of resilience that has gradually waned with the persistence of the emergency and the accumulating social and economic effects of the pandemic [34]. 

The current study compared different groups of patients diagnosed with migraine before and during the COVID-19 pandemic, which did not allow us to assess the development of individual patients over time. A further limitation is the difficulty in determining a causal link between the occurrence of the pandemic and changes in chronic pain. Moreover, patients’ history of chronic pain prior to the study period is unknown. Another limitation of the study relates to the numbers of subjects in inhomogeneous groups.

## 5. Conclusions

The prolonged pandemic had a negative impact on the evolution of migraines. During the second pandemic wave, symptoms of anxiety and depression increased. The return to school can be considered a stress factor for adolescents with migraine, and it is also important to consider the impact of COVID-19 restrictions and their prolongation over time.

These aspects suggest the importance of carefully considering the emotional state of migraine patients.

## Figures and Tables

**Figure 1 brainsci-13-00273-f001:**
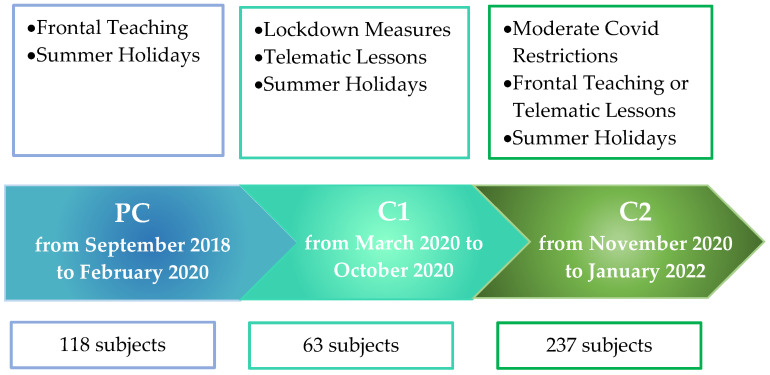
Timeline and periods description. PC: Pre COVID; C1: COVID 1; C2: COVID 2.

**Figure 2 brainsci-13-00273-f002:**
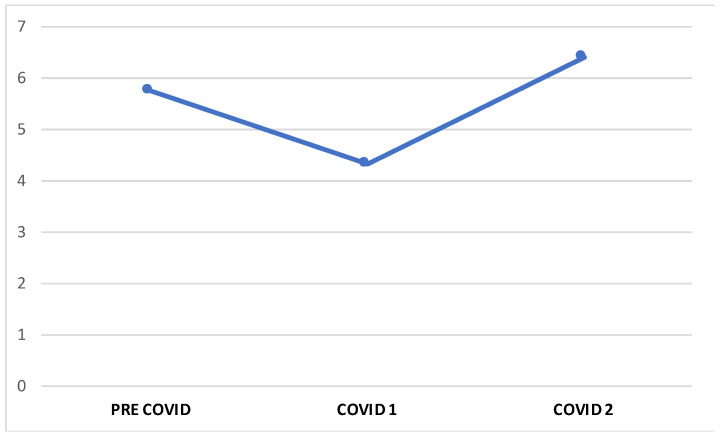
Average number of migraine episodes in the ore-COVID and COVID-19 periods.

**Figure 3 brainsci-13-00273-f003:**
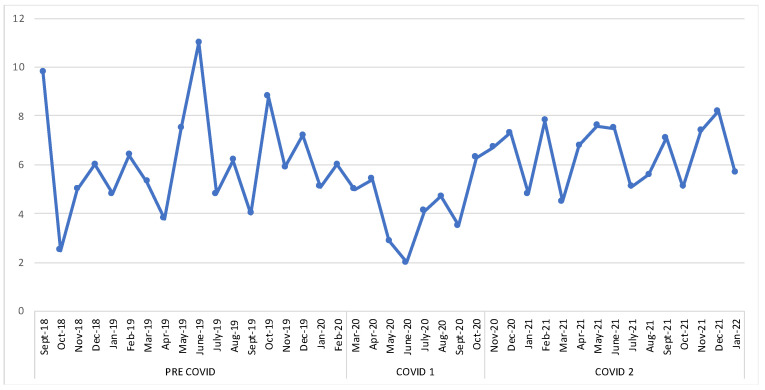
Average number of migraine episodes over the time span.

**Figure 4 brainsci-13-00273-f004:**
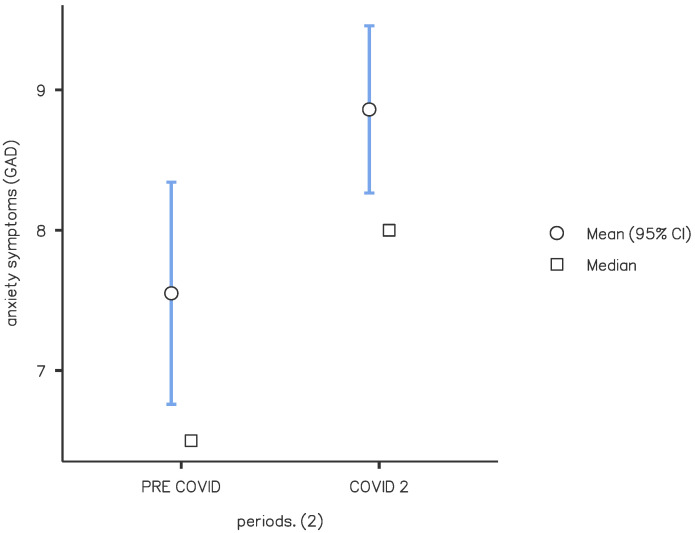
Anxiety symptoms: comparison of pre-COVID and COVID 2 periods. Mean score of the GAD-7 questionnaire for the screening of anxiety symptoms. Significance between pre-COVID and COVID 2 groups (*p* = 0.012).

**Figure 5 brainsci-13-00273-f005:**
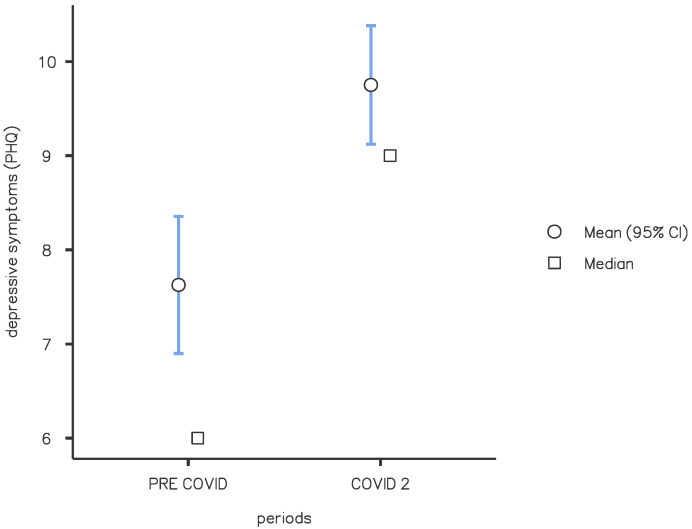
Depression symptoms: comparison of pre-COVID and COVID 2 periods. Mean score of the PHQ-9 questionnaire for the screening of depression symptoms. Significance between pre-COVID and COVID 2 groups (*p* < 0.001).

**Table 1 brainsci-13-00273-t001:** Distribution of patients according to frequency, intensity of attacks, and use of prophylaxis in the general sample and in the different periods. SD: standard deviation.

	Total Patients N 418	High Frequency N 206 (49%)	Low Frequency N 211 (51%)	Severe Intensity N 166 (40%)	Mild Intensity N 252 (60%)	Ongoing Prophylaxis Treatment N 169 (40%)	No Prophylaxis Treatment N 249 (60%)
Demographic							
features	Mean (SD)	Mean (SD)	Mean (SD)	Mean (SD)	Mean (SD)	Mean (SD)	Mean (SD)
Age in years	14.2 (1.71)	14.4 (1.74)	13.9 (1.65)	14.3 (1.75)	14.1 (1.68)	14.4 (1.75)	14.1 (1.68)
Sex	Number (%)	Number (%)	Number (%)	Number (%)	Number (%)	Number (%)	Number (%)
Females	308 (74)	167 (81)	141 (67)	131 (43)	177 (57)	113 (37)	195 (63)
Males	110 (26)	39 (19)	71 (33)	36 (33)	74 (67)	33 (30)	77 (70)
Periods	Number (%)	Number (%)	Number (%)	Number (%)	Number (%)	Number (%)	Number (%)
Pre COVID	118	53 (45)	65 (55)	46 (39)	72 (61)	31 (26)	87 (74)
School period	98	45 (46)	53 (54)	41 (42)	57 (58)	24 (24)	74 (76)
Summer period	20	8 (40)	12 (60)	5 (25)	15 (75)	7 (35)	13 (65)
COVID	300	153 (51)	147 (49)	121 (40)	179 (60)	115 (38)	185 (62)
School period	220	130 (56)	130 (56)	99 (42)	134 (58)	97 (42)	136 (58)
Summer period	80	23 (34)	44 (66)	22 (33)	45 (67)	18 (27)	49 (73)
COVID 1	63	20 (32)	43 (68)	20 (32)	43 (68)	16 (25)	47 (65)
School period	36	11 (31)	25 (69)	13 (36)	23 (64)	13 (36)	23 (64)
Summer period	27	9 (33)	18 (67)	7 (26)	20 (74)	3 (11)	24 (89)
COVID 2	237	133 (56)	104 (44)	100 (42)	136 (58)	99 (42)	138 (58)
School period	197	119 (60)	78 (49)	86 (44)	111 (56)	84 (43)	113 (57)
Summer period	40	14 (35)	26 (65)	15 (37)	25 (63)	15 (37)	25 (63)

## Data Availability

Data available on request due to privacy restrictions.

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
