# Peer review of "Long-Term Effects of COVID-19 Pandemic on Migraine in Adolescents. A Retrospective Analysis of the Population Attending the Headache Center in Different Phases of the Pandemic"

_brainsci, 2023, doi:10.3390/brainsci13020273_

Round 1

Reviewer 1 Report

Question 1: The patients were classified into the HF group and LF group according to their frequency. What is the basis for the distinction of 4 episodes per month?

Question 2: Whether the part of lines 190 to 205 is repeated?

Author Response

Question 1: The patients were classified into the HF group and LF group according to their frequency. What is the basis for the distinction of 4 episodes per month?

a cut-off of 4 attacks per month was established because this is generally the frequency above which to consider starting migraine prophylaxis in childhood.

Papetti L, Ursitti F, Moavero R, Ferilli MAN, Sforza G, Tarantino S, Vigevano F, Valeriani M. Prophylactic Treatment of Pediatric Migraine: Is There Anything New in the Last Decade? Front Neurol. 2019 Jul 16;10:771. doi: 10.3389/fneur.2019.00771.

Maryam Oskoui 1, Tamara Pringsheim 1, Lori Billinghurst 1, Sonja Potrebic 1, Elaine M Gersz 1, David Gloss 1, Yolanda Holler-Managan 1, Emily Leininger 1, Nicole Licking 1, Kenneth Mack 1, Scott W Powers 1, Michael Sowell 1, M Cristina Victorio 1, Marcy Yonker 1, Heather Zanitsch 1, Andrew D Hershey 1.Practice guideline update summary: Pharmacologic treatment for pediatric migraine prevention: Report of the Guideline Development, Dissemination, and Implementation Subcommittee of the American Academy of Neurology and the American Headache Society Ref. Neurology 2019 Sep 10;93(11):500-509. doi: 10.1212/WNL.0000000000008105. Epub 2019 Aug 14.

Question 2: Whether the part of lines 190 to 205 is repeated?

We apologize, we have deleted the repetition of the text.

Reviewer 2 Report

The article "Long-term effects of Covid-19 Pandemic on migraine in adolescents.A retrospective analysis of the population belonging to the headache center in different phases of the pandemic" contains very interesting information and can be helpful in the management of migraine patients in future during such pandamic. The authors have made good efforts but few places are there where improvement can enhance the quality of article.

1. Remove full stop after the title of manuscript

2. There is no mention of "the to study the relationship between characteristics of headache episodes and 77 psychological and school-related aspects." in the abstract.

3. Quality/visibility of figures can be improved.

4. Please check the format of the tables given in the article.

5. Should the title of table 1 be not above the table?

6. References needs a comprehensive formatting to harmonizing with journal's format.

Author Response

  1. Remove full stop after the title of manuscript

Thank you. We have corrected this error.

  1. There is no mention of "the to study the relationship between characteristics of headache episodes and 77 psychological and school-related aspects." in the abstract.

We added in the abstract this aim.

  1. Quality/visibility of figures can be improved.

Thank you for the indication. We have improved the quality/visibility of the figures.

  1. Please check the format of the tables given in the article.

We apologize, the format was not suitable. We have changed the format of the table.

  1. Should the title of table 1 be not above the table?

Yes. We have moved the title of the table

  1. References needs a comprehensive formatting to harmonizing with journal's format.

Thank you. We have formatted the bibliographic references according to the journal guidelines.